# Subjective HRQoL in Patients with Sleep Apnea Syndrome Who Underwent PAP Therapy in a Rehabilitation Setting: A Longitudinal Study

**DOI:** 10.3390/jcm12051907

**Published:** 2023-02-28

**Authors:** Antonia Pierobon, Martina Vigorè, Eugenia Taurino, Gemma Grassi, Valeria Torlaschi, Marina Maffoni, Rita Maestroni, Roberto Maestri, Francesco Fanfulla

**Affiliations:** 1Psychology Unit of Montescano Institute, Istituti Clinici Scientifici Maugeri IRCCS, 27040 Montescano, Italy; 2Respiratory Function and Sleep Medicine Unit, Istituti Clinici Scientifici Maugeri IRCCS, 27040 Montescano, Italy; 3Pulmonary Rehabilitation Division of Montescano Institute, Istituti Clinici Scientifici Maugeri IRCCS, 27040 Montescano, Italy; 4Department of Biomedical Engineering of Montescano Institute, Istituti Clinici Scientifici Maugeri IRCCS, 27040 Montescano, Italy

**Keywords:** OSA, PAP therapy, HRQoL, emotional status, interdisciplinary rehabilitation

## Abstract

Background: Obstructive sleep apnea (OSA) is often associated with decreased health-related quality of life (HRQoL). The aims of this study were to evaluate HRQoL, the clinical and psychological profile of suspected or verified OSA patients, and the impact of PAP therapy at 1-year follow-up. Methods: At T0, OSA-suspected subjects underwent clinical, HRQoL, and psychological assessment. At T1, OSA patients underwent PAP therapy in a multidisciplinary rehabilitation setting. At 1 year follow-up, OSA patients were evaluated for the second time. Results: At T0, OSA patients (n = 283) and suspected OSA subjects (n = 187) differed for AHI, BMI, and ESS. At T0, the PAP-treatment group (n = 101) showed moderate–severe anxious (18.7%) and depressive (11.9%) symptoms. At 1 year follow-up (n = 59), the sleep breathing pattern had normalized and there was a reduction of ESS scores and anxious symptoms. There was also an improvement in HRQoL (0.6 ± 0.4 vs. 0.7 ± 0.5, *p* = 0.032; 70.4 ± 19.0 vs. 79.2 ± 20.3, *p* = 0.001) and in satisfaction with sleep quantity (52.3 ± 31.7 vs. 71.4 ± 26.2, *p* = 0.001), sleep quality (48.1 ± 29.7 vs. 70.9 ± 27.1, *p* = 0.001), mood (58.5 ± 24.9 vs. 71.0 ± 25.6, *p* = 0.001), and physical resistance (61.6 ± 28.4 vs. 67.8 ± 27.4, *p* = 0.039). Conclusion: Considering the impact of PAP treatment on patients’ psychological and HRQoL evaluations that we observed, our data are valuable for unveiling different profiles characterizing this clinical population.

## 1. Introduction

Obstructive sleep apnoa Syndrome (OSA) is a common chronic disease characterized by apneas or hypopneas during sleep due to an upper airway collapse [1] and is associated with intermittent hypoxia and arousal from sleep. This condition can lead to daytime symptoms such as excessive sleepiness, fatigue, and lack of concentration [2]. If not adequately treated, this disorder can negatively impact both physical and psychological health, deteriorating the individual’s daily and working life and reducing health-related quality of life (HRQoL) [2].

A consistent number of studies have highlighted the importance of HRQoL and psychological and neuropsychological assessment among patients affected by OSA and undergoing positive airway pressure (PAP) therapy. The improvement of daytime symptoms and HRQoL is considered one of the goals of OSA therapy. A systematic review and meta-analysis demonstrated that PAP therapy improves HRQoL, with a positive impact on aspects of physical and mental health [2]. Specifically, the following different areas were investigated: psychopathological aspects (anxiety and/or depression), HRQoL, adherence to treatment, and cognitive domains [1,3,4,5,6,7].

It is also crucial to report the possible link between PAP therapy and psychopathological symptoms; depressive ones in particular. Indeed, previous research has described the negative impact of depression on the adherence to PAP therapy and the efficacy of PAP therapy not only in reducing OSA symptoms and comorbidities but also depressive symptoms [8,9,10,11]. The difficult interpretation of this relationship can be due to the overlapping of OSA and mood symptoms (e.g., sleepiness, reduction of physical exercise, reduction of usual activities and interests) [11].

Despite the growing interest in the evaluation of HRQoL and psychological symptoms in OSA patients, there is still little research about these aspects in the rehabilitative context. For instance, a previous study investigated these issues in OSA patients with cardiovascular comorbidities [12].

To the best of our knowledge, an evaluation of subjective and holistic well-being is still lacking for OSA patients and subjects suspected of OSA [13]. Thus, further studies are needed. Longitudinal observations are particularly needed, as they lead to a better understanding of the impact of OSA over time.

Therefore, this study attempted to provide an overall evaluation of HRQoL and psychological factors over time and after ventilatory therapy. Specifically, the aims of this longitudinal prospective study were as follows:To provide an overall evaluation of the psychological profile (i.e., anxious and depressive symptoms, HRQoL at the diagnostic phase) of OSA patients versus subjects with suspected, but not confirmed, OSA.To evaluate psychological symptoms, HRQoL, subjective impact, and self-management of PAP therapy at 1-year follow-up in OSA patients hospitalized in a multidisciplinary rehabilitation setting.

## 2. Materials and Methods

### 2.1. Subjects

We enrolled treatment-naïve OSA patients according to ICSD-3 diagnostic criteria (mild obstructive sleep apnea: AHI 5–14 per h, moderate obstructive sleep apnea: AHI 15–29 per h, and severe obstructive sleep apnea: AHI ≥ 30 per h), consecutively admitted to the Sleep Unit of Istituti Clinici Scientifici Maugeri IRCCS of Montescano Institute for suspected OSA.

This research project was accepted by the Central Ethical Committee of Clinical Scientific Institutes Maugeri IRCCS (Protocol n° 2026, 03/03/2016).

The exclusion criteria were the following: older than 75 years, severe clinical issues (e.g., severe CHF, respiratory failure, etc.), cognitive and/or psychiatric co-morbidities precluding evaluation, and no Italian education or relapse into illiteracy.

Eligible patients were informed in speech and writing about the scope of the study, and they were asked to sign an informed consent. The research was performed on a voluntary basis during clinical practice and without any reimbursement for the participants.

### 2.2. Procedure

All patients undergoing a clinical sleep evaluation for suspected OSA were asked to participate (T0).

Patients with polysomnographic evidence of OSA were admitted to hospital for an interdisciplinary rehabilitation program (3–4 weeks) (T1), which included educational sessions, adaption to and titration of PAP, exercise training (cycloergometer and/or treadmill, arms ergometer), inspiratory muscles training, psychological intervention, and metabolic evaluation with a personalized diet according to clinical recommendations. PAP titration was performed during a full standard polysomnography (PSG) according to AASM statement. PAP therapy was started during the hospital stay and adherence was checked daily by technicians; specific interventions were properly adopted, such as changes in masks or level of humidification. This multidisciplinary clinical and rehabilitative care path was also previously described in the literature [7].

After discharge, patients were re-evaluated after 1-year follow-up (T2).

At T0, the assessment was composed of a socio-demographic schedule, the Hospital Anxiety and Depression Scale (HADS) [14,15], EQ-5D-3L—VAS [16,17], and the 8-Item Satisfaction Profile (SAT-P) [18]. After adaptation to PAP therapy (T1), lasting approximately two or three weeks, the HRQoL evaluation (EQ VAS) was repeated. At T2, 2 items of SAT-P were added to the baseline evaluation.

### 2.3. Instruments

Socio-demographic schedule investigates socio-anagraphic variables (e.g., age, gender, marital status, caregiver) and comorbidities, as well as some clinical data and risk factors (e.g., body mass index (BMI) and smoking habit).

Subjective sleepiness was assessed by the Epworth Sleepiness Scale (ESS) [19,20]. The responder is asked to rate on a 4-point scale how likely they are to doze or fall asleep while engaging in 8 typical everyday activities in order to assess the level of daytime sleepiness.

The Hospital Anxiety and Depression Scale (HADS) is a scale that identifies the presence of anxiety and depressive symptoms in hospitalized patients [14,15]. It is made up of 14 items divided into two subscales (7 items each) that measure depressive and anxious symptoms, respectively. Each item ranges from 0 to 3. Through the sum of the responses, two distinct scores are obtained, one for anxious symptoms and one for depressive ones. This scale also enables a severity assessment of symptoms based on the score obtained: 0–7, normal; 8–10, mild; 11–14, moderate; 15–21, serious.

The EQ-5D-3L is a generic quality-of-life measurement tool. It investigates five dimensions: mobility, self-care, usual activities, pain/discomfort, and anxiety/depression. Each item has three levels: 1: no problem; 2: some problems; 3: extreme problems. The second section consists of a VAS scale from 0 (worst imaginable health status) to 100 (best imaginable health status), and the subject is asked to indicate their level of self-perceived wellness [16,17]. In this study, EQ-VAS was used to stratify the sample at T0 (G1: EQ-VAS ≤ 50; G2: 50 < EQ-VAS ≥ 75; G3: EQ-VAS > 75).

The 8-Item Satisfaction Profile (SAT-P) produces a visual representation of satisfaction level on a 10 cm line that represents the lowest and highest degrees of satisfaction possible. The patient is asked to indicate with a sign on the line the degree of satisfaction perceived [18]. The investigated areas were sleep quantity; sleep quality; mood; resistance to physical fatigue; mental efficiency; partner relationship; friends relationship; colleagues relationship. At T2, self-management of PAP therapy and impact of PAP therapy were also investigated.

### 2.4. Statistical Analysis

Descriptive statistics were reported as mean ± standard deviation for continuous variables and as N or percent frequency for discrete variables. Between-group comparisons (OSA vs. No OSA) for continuous variables were carried out via Mann–Whitney U test. Within-group comparisons (T0 vs. T2) for continuous variables were carried out via Wilcoxon signed-rank test.

Within-group comparisons for three observation times (T0, T1, and T2) for continuous variables were carried out by a Friedman’s test (non-parametric version of repeated measures analysis of variance). A significant result was followed up with post-hoc analysis (Dunn–Sidak criterion) to compare values at T0 vs. T1, T0 vs. T2, and T1 vs. T2.

The association between psychological variables, HRQoL and adherence to PAP treatment at 1-year follow-up was assessed by correlation analysis (Spearman’s correlation coefficient).

All statistical tests were two-tailed, and a *p*-value <0.05 was considered statistically significant. When appropriate, multiple comparison procedures were used, controlling the false discovery rate (FDR) at 5% using the Benjamini–Hochberg method. All analyses were carried out using the SAS/STAT statistical package, release 9.4 (SAS Institute Inc., Cary, NC, USA).

## 3. Results

Figure 1 shows the flow diagram of the research timing and the division of patients according to diagnosis and treatment carried out.

At T0, 243 OSA suspicion subjects were enrolled; of these, 187 (mean age 55.9 ± 10.6, 40 females and 147 males) were diagnosed as OSA, while 56 were not OSA patients (53.1 ± 9.7, 22 female and 34 males).

Of the 187 OSA patients, 86 (55.7 ± 10.8, 17 females and 69 males) did not undergo PAP therapy. Specifically, 34 patients dropped out, 30 subjects underwent outpatient clinical monitoring with specialist-provided behavioral recommendations, and 22 patients were admitted to hospital for a multidisciplinary rehabilitative intervention without PAP adaptation because it was not recommended.

At T1, 101 OSA patients (56.3 ± 10.5, 24 females and 77 males) were admitted to Istituti Clinici Scientifici Maugeri IRCCS for adaptation to PAP therapy and the multidisciplinary rehabilitation. Of these, at T2, 59 OSA patients (mean age 58.7 ± 10.4, 11 females and 48 males) were readmitted to hospital for monitoring the PAP treatment and carrying out a second multidisciplinary rehabilitative intervention, while 42 patients dropped out (Figure 1).

### 3.1. Description and Comparison between OSA and No OSA Outpatients at Baseline (n = 187 vs. n = 56)

As reported in Table 1, upon comparing outpatients with confirmed and unconfirmed OSA diagnosis, significant differences were found regarding clinical variables in AHI (37.5 ± 23.1 vs. 9.8 ± 7.1, *p* = 0.001), BMI (31.1 ± 6.6 vs. 27.7 ± 5.7, *p* = 0.001), and EES (5.6 ± 3.8 vs. 4.3 ± 3.5, *p* = 0.025), which were higher in OSA patients. Between these groups, no significant differences were found concerning anxious and depressive symptoms, HRQoL, or 8-Item SAT-P.

### 3.2. Description and Comparison between Baseline and 1-Year Follow-Up in PAP-Treatment Patients (n = 59)

At T0, the inpatients PAP-Tr group showed moderate to severe anxious (18.7%) and depressive (11.9%) symptoms, respectively.

At 1-year follow-up, 86.4% of patients were considered adherent to treatment using PAP therapy (mean usage at least 4 h/night for/>70% of nights). These data refer to patients for whom PAP use was available (n = 44).

As reported in Table 2, the sleep breathing pattern recorded at baseline had normalized with PAP treatment at 1-year follow-up, and we observed a significant reduction in ESS score (5.9 ± 3.9 vs. 2.4 ± 2.6, *p* = 0.001). Similarly, there was a significant reduction in anxious symptoms (6.8 ± 3.6 vs. 5.4 ± 3.8, *p* = 0.001). There was also a significant improvement in HRQoL (0.6 ± 0.4 vs. 0.7 ± 0.5, *p* = 0.032; 70.4 ± 19.0 vs. 79.2 ± 20.3, *p* = 0.001), such as in satisfaction with sleep quantity (52.3 ± 31.7 vs. 71.4 ± 26.2, *p* = 0.001), sleep quality (48.1 ± 29.7 vs. 70.9 ± 27.1, *p* = 0.001), mood (58.5 ± 24.9 vs. 71.0 ± 25.6, *p* = 0.001), physical resistance (61.6 ± 28.4 vs. 67.8 ± 27.4, *p* = 0.039), and colleagues relationship (68.5 ± 23.1 vs. 76.8 ± 21.5, *p* = 0.027).

### 3.3. Comparison of HRQoL at T0, T1, and T2

Among patients with EQ VAS scores available at T0, T1, and T2 (N = 47), the corresponding scores were 73.6 ± 16.8, 84.8 ± 15.2, and 82.1 ± 18.2. Friedman’s analysis revealed a significant time effect (*p* = 0.0002), and post hoc analysis showed that EQ VAS scores significantly improved at T1 (*p* = 0.0008, T1 vs. T0) and was maintained at T2 (*p* = 0.002, T2 vs. T0). Conversely, no difference between T1 and T2 was observed (*p* = 0.97).

We divided our patients into two groups according to their AHI severity baseline (AHI, T_0_ ≤ 30 = 22 (10.3 ± 16.4) and AHI, T_0_ ≥ 30 = 37 (7.9 ± 22.2)), and then we correlated AHI severity scores with Δ(T_2_ − T_0_) EQ VAS. No significant correlation was found (*p* = 0.96) between OSA severity and HRQoL.

Furthermore, according to various degrees of perceived HRQoL at enrollment (G1-G3), a different pattern of improvement in EQ-VAS at T2, was found, as shown in Figure 2: the lower the level of HRQoL at the baseline, the higher the degree of improvement at the follow-up (r = −0.47, *p* = 0.0002; Figure 2).

## 4. Discussion

The present longitudinal study investigated the psychological profile of OSA patients (i.e., anxious and depressive symptoms, HRQoL, and subjective satisfaction) at the diagnostic phase, making comparisons with subjects with suspected but not confirmed OSA. Moreover, the subjective impact and the self-management of PAP therapy at 1-year follow-up in a rehabilitation setting and its correlation with psychological variables were also analyzed. Our results are described in detail below.

Firstly, at baseline, no differences in psychological, HRQoL, or satisfaction profile were found between OSA patients and suspicion OSA subjects, while the two groups differed for AHI, BMI, and ESS. Our data appear to be in line with Isodoro et al. on the absence of the relation between HRQoL perception among subjects referred for OSA and disease severity [21]. As for psychological symptoms, our results are consistent with Gharsalli et al. (2022), as they also found no correlations between HADS scores and AHI [22]. However, further studies are necessary to better understand this finding. Concerning BMI, our values (31.1 ± 6.6 vs. 27.7 ± 5.7, *p* = 0.001) are consistent with those found by Ersözlü et al. (34.4 ± 5.9 vs. 29.4 ± 4.0) [23] in patients with similar AHI. Between groups, there was a significant difference in sleepiness symptoms (5.6 ± 3.8 vs. 4.3 ± 3.5, *p* = 0.008) according to ESS scores, even if both groups were identified as not sleepy differently from the majority of the existing relevant literature [24,25].

Almost a quarter of OSA patients who underwent PAP treatment showed, at baseline, moderate to severe anxious and depressive symptoms. These data are similar to those found by Morrone et al. regarding anxious symptoms, even if depressive symptoms were higher in our sample [26]. However, the current findings differed from a previous meta-analysis describing the pooled prevalence of depressive and anxious symptoms in OSA patients at 35.0% and 32.0%, respectively [27].

Secondly, regarding the comparison between baseline and 1-year follow-up, sleep breathing patterns had normalized after PAP treatment (41.8 ± 20.6 vs. 2.7 ± 2.8, *p* = 0.001), and we observed a significant reduction in subclinical ESS scores (5.9 ± 3.9 vs. 2.4 ± 2.6, *p* = 0.001). These data are consistent with the existing literature on the efficacy of the PAP treatment for reducing clinical and sleepiness symptoms, even if patients in this study were not sleepy [28,29,30,31]. In addition, there was a significant reduction of anxious symptoms (6.8 ± 3.6 vs. 5.4 ± 3.8, *p* = 0.001) and a significant improvement in HRQoL (0.6 ± 0.4 vs. 0.7 ± 0.5, *p* = 0.032; 70.4 ± 19.0 vs. 79.2 ± 20.3, *p* = 0.001). An improvement of HRQoL was already reported in a previous study, with OSA patients undergoing the same multidisciplinary treatment [7]. Overall, these results find extensive confirmation in the worldwide literature [2,27,32], and they underline the pivotal role that the evaluation of these variables has in the management of OSA disease. Besides the aforementioned enhancements, the perceived subjective satisfaction reported by OSA patients regarding sleep quantity (52.3 ± 31.7 vs. 71.4 ± 26.2, *p* = 0.001), sleep quality (48.1 ± 29.7 vs. 70.9 ± 27.1, *p* = 0.001), mood (58.5 ± 24.9 vs. 71.0 ± 25.6, *p* = 0.001), physical resistance (61.6 ± 28.4 vs. 67.8 ± 27.4, *p* = 0.039), and colleagues relationship (68.5 ± 23.1 vs. 76.8 ± 21.5, *p* = 0.027) were higher after 1 year of PAP therapy.

Furthermore, a significant improvement emerged in HRQoL comparing EQ VAS scores at T0 and T1 (73.6 ± 16.8 vs. 84.8 ± 15.2, *p* < 0.0001), and, as aforementioned, between T0 and T2 (73.6 ± 16.8, vs. 82.1 ± 18.2, *p* = 0.012). The positive impact that PAP therapy might have on HRQoL is well-known in the literature [2,24,33], and the absence of differences in EQ VAS between T1 and T2 (84.8 ± 15.2 vs. 82.1 ± 18.2, *p* = 0.47) could indicate the maintenance of the positive effect over time, specifically after one year from rehabilitative hospitalization. However, the absence of a control group without PAP treatment prevents assigning the betterment of HRQoL only to multidisciplinary rehabilitation and PAP. These data were consistent with Lo Bue and colleagues (2020), who reported a significant improvement in HRQoL from a minimum of 2 to a maximum of 9 months of PAP therapy follow-up [32]. Thus, our data enrich the knowledge in the field, showing that the HRQoL improvement could last over time, specifically up to 12 months of PAP therapy.

To conclude, we must note that this study was not without limitations. Firstly, the absence of a control group prevents making conclusions on the efficacy of the treatment itself. Secondly, the small size of the sample does not allow for generalization of the findings; thus, more studies involving wider samples are to be welcomed. In addition, subjects and patients were enrolled in one single place in northern of Italy and were not old; thus, socio-cultural biases might exist.

Besides these limits, some merits deserve to be highlighted as well. First of all, to our knowledge, there is scant research analyzing adherence to PAP treatment in in-patients undergoing a multidisciplinary rehabilitation program. Specifically, this study provided a sound example of a holistic care-taking of the patient and put attention on subjective quality of life and anxiety and depressive symptoms in an attempt to gain a more subjective perspective of this clinical condition. This study also took into consideration patients suspected of OSA, unveiling possible differences with patients having a clear diagnosis and, in turn, providing information that is useful for tailored interventions. Indeed, psychological and HRQoL evaluations in suspected OSA subjects and PAP-treatment patients might offer valuable data for unveiling different profiles of this clinical population from a longitudinal perspective.

## Figures and Tables

**Figure 1 jcm-12-01907-f001:**
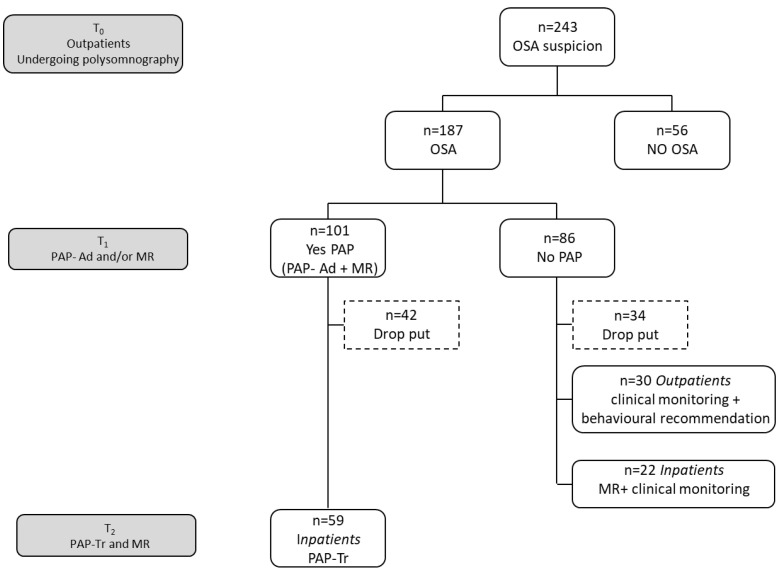
Flow diagram of research timing and subdivision of subjects according to diagnosis and treatment. Abbreviations: PAP-Ad, PAP adaptation; MR, multidisciplinary rehabilitation; PAP-Tr, PAP treatment.

**Figure 2 jcm-12-01907-f002:**
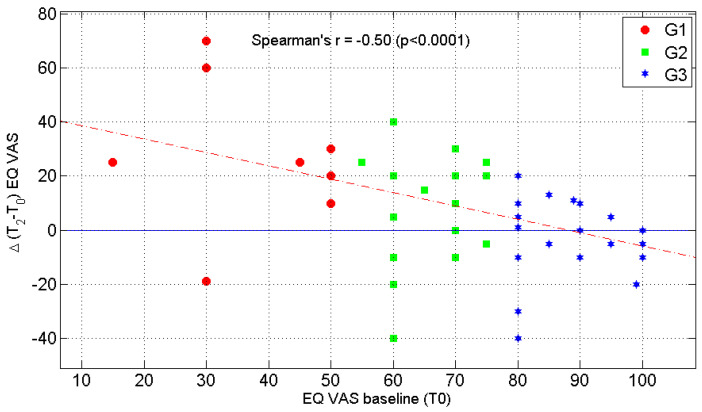
Association between Δ(T2 − T0) EQ VAS and baseline EQ-VAS (T0) according to sample stratification at T0 G1: EQ-VAS ≤ 50; G2: 50 < EQ-VAS ≤ 75; G3: EQ-VAS > 75.

**Table 1 jcm-12-01907-t001:** Comparison of clinical, psychological, and HRQoL data of outpatients undergoing polysomnography (T0).

	Outpatients (m ± SD)		
Variables	Yes OSA n = 187	No OSA n = 56	*p*
AHI	37.7 ± 23.2	9.8 ± 7.1	<0.0001
BMI	31.2 ± 6.7	27.6 ± 5.7	<0.0001
ESS	5.6 ± 3.8	4.3 ± 3.5	0.008
Age	56.0 ± 10.6	52.9 ± 9.8	0.054
HADS			
Anxiety	5.7 ± 3.8	5.9 ± 4.0	0.71
Depression	4.3 ± 3.3	4.4 ± 3.5	0.80
EQ Index	0.7 ± 0.4	0.7 ± 0.3	0.64
EQ VAS	72.5 ± 18.2	70.4 ± 18.3	0.48
SAT-P			
Sleep quantity	58.0 ± 30.6	58.9 ± 32.2	0.80
Sleep quality	54.0 ± 30.5	53.2 ± 31.8	0.93
Mood	65.2 ± 26.0	63.7 ± 26.6	0.67
Resistance to physical fatigue	59.5 ± 30.1	64.3 ± 28.9	0.31
Mental efficiency	75.7 ± 24.7	76.5 ± 23.2	0.92
Partner relationship	70.8 ± 29.4	72.6 ± 25.4	0.97
Friends relationship	75.3 ± 24.5	69.6 ± 28.7	0.23
Colleagues relationship	70.8 ± 25.8	73.3 ± 21.4	0.98

Abbreviations: AHI, Apnea–Hypopnea Index; BMI, body mass index; ESS, Epworth Sleepiness Scale; HADS, Hospital Anxiety and Depression Scale, and SAT-P, Satisfaction Profile. Note: All significant comparisons were confirmed while controlling the false discovery rate at 5%.

**Table 2 jcm-12-01907-t002:** Comparison of clinical, psychological, and HRQoL data of OSA patients at baseline and 1 year of PAP treatment (n = 59).

	PAP-Tr Patients (m ± SD)		
Variables	T_0_	T_2_	*p*
AHI	41.8 ± 20.6	2.7 ± 2.8	<0.0001
BMI	30.8 ± 5.0	30.8 ± 4.6	1.00
ESS	5.9 ± 4.0	2.4 ± 2.6	<0.0001
HADS			
Anxiety	6.8 ± 3.6	5.4 ± 3.8	0.001
Depression	5.3 ± 3.5	4.8 ± 4.1	0.14
EQ Index	0.61 ± 0.43	0.65 ± 0.49	0.032
EQ VAS	70.4 ± 19.0	79.2 ± 20.3	0.0007
SAT-P			
Sleep quantity	52.3 ± 31.7	71.4 ± 26.2	<0.0001
Sleep quality	48.1 ± 29.7	70.8 ± 27.1	<0.0001
Mood	58.5 ± 24.9	71.0 ± 25.6	0.00018
Resistance to physical fatigue	61.6 ± 28.4	67.8 ± 27.4	0.039
Mental efficiency	74.4 ± 23.6	78.2 ± 24.4	0.14
Partner relationship	72.3 ± 24.4	73.3 ± 23.2	0.47
Friends relationship	76.4 ± 20.2	78.7 ± 23.4	0.26
Colleagues relationship	68.5 ± 23.1	76.8 ± 21.5	0.027
Subjective impact of PAP-Tr	-	81.7 ± 21.8	-
Self-management of PAP-Tr	-	82.5 ± 21.6	-

Abreviations: PAP-Tr, PAP treatment; AHI, Apnea–Hypopnea Index; BMI, body mass index; ESS, Epworth Sleepiness Scale; HADS, Hospital Anxiety and Depression Scale SAT-P, Satisfaction Profile. Note: All significant comparisons, except SAT-P Resistance to physical fatigue, were confirmed while controlling the false discovery rate at 5%.

## Data Availability

Inquiries can be directed to the corresponding author.

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
