# Peer review of "Subjective HRQoL in Patients with Sleep Apnea Syndrome Who Underwent PAP Therapy in a Rehabilitation Setting: A Longitudinal Study"

_jcm, 2023, doi:10.3390/jcm12051907_

Round 1

Reviewer 1 Report

In this study, Antonia et al conducted a longitudinal study of PAP therapy in patients with obstructive sleep apnea to determine whether PAP therapy improved healthcare related quality of life (HRQoL) over a 1 year follow up. At the end of 1 year, they observed the PAP improved and sustained HRQoL as well as sleep quantity, quality mood and physical resistance. There are limited data on the impact of PAP over this length of time and thus the data are of interest. However, there are concerns that need to be addressed.

1.     Of the 243 patients originally enrolled, 42 dropped out in the PAP group and 34 in those who did not receive PAP. Why did these participants drop out? Furthermore, was there any attempt to determine the HRQoL and sleep characteristics in these patients and compare them to those who used PAP?

2.     In those patients who received PAP, what was their hourly adherence? Most PAP devices now have incorporated monitoring of hourly PAP usage. Was there a correlation between PAP usage and improvement in HRQoL?

3.     Was there an association between HRQoL and severity of OSA or degree of sleepiness?

4.     You observed improvements in anxiety and possible depression. Can the improvement in HRQoL be explained at least partially by these factors?

5.     Regarding your statistical analysis, you conducted between group comparisons using the Mann-Whitney U test and within group comparisons with the Wilcoxon signed-rank test. These procedures are generally used when your data is not normally distributed. Were your data significantly skewed in one direction or the other? If they were not normally distributed, then use of a repeated measures ANOVA is not valid because the ANOVA is not used for non-parametric data. The appropriate test would be a Friedman’s non parametric ANOVA. Similarly, correlations would need to be performed with a Spearman’s rho.

6.     Your limitations paragraph needs to mention the potential impact of dropouts biasing the data towards only persons who benefited from PAP.

Minor Issues:

1.     Line 1: “systemic review” should be “systematic review”

2.     Lion 84: “polysonoraphic” should be “polysomnographic”

3.     What was your definition of OSA? AHI ≥5,  ≥10 or ?

4.     What was your definition of hypopnea when defining the AHI? You said that you used the AASM definition, but there are 2 definitions of hypopnea.

5.     In your discussion, you should cite the CATNAP study which was a RCT showing 8 weeks of CPAP improved the FOSQ.

a.     Weaver TE, Mancini C, Maislin G, Cater J, Staley B, Landis JR, Ferguson KA, George CF, Schulman DA, Greenberg H, Rapoport DM, Walsleben JA, Lee-Chiong T, Gurubhagavatula I, Kuna ST. Continuous positive airway pressure treatment of sleepy patients with milder obstructive sleep apnea: results of the CPAP Apnea Trial North American Program (CATNAP) randomized clinical trial. Am J Respir Crit Care Med. 2012 Oct 1;186(7):677-83. doi: 10.1164/rccm.201202-0200OC. Epub 2012 Jul 26. PMID: 22837377; PMCID: PMC3480519.

Reviewer 2 Report

This an interesting longitudinal study assessing symptoms in people with OSA after one year of using CPAP. As it is not a randomized controlled study definitive conclusions about CPAP cannot be made, but this study does show that for some outcomes variables such as AHI and ESS, the effects of CPAP are seen for one year.

The reasons for the exclusion criteria are not clear to me. Why exclude people over 75 years? Why should people with comorbid conditions be excluded as these are not uncommon in people with OSA and so need to be studies as well? It does make sense to exclude people with psychiatric conditions when the main purpose of the study is to assess the effects of CPAP on psychological symptoms.

What was definition of OSA in this study, for example, AHI, oxygen desaturation, arousal index?

For how long were patients in the hospital for the rehabilitation programme? How was it decided who would have second rehabilitation programme? How long was there between the two admissions?

If those without OSA or with minimal OSA had the same degree of psychological symptoms as those with more significant OSA, is it possible to conclude that the symptoms were due to OSA?

Do you have any data on adherence with CPAP?

Round 2

Reviewer 1 Report

The authors did not make the changes in statistical analysis that they purported to have done in their author response.

Author Response

The authors did not make the changes in statistical analysis that they purported to have done in their author response.

Responce. Thank you for the punctual observation, there was a little mistake in uploading the files. We have modified the as reported below in the "Statistical analysis" paragraph:

"Descriptive statistics were reported as mean ± standard deviation for continuous variables and as N or percent frequency for discrete variables. Between-group comparisons (OSA vs No OSA) for continuous variables were carried out by the Mann-Whitney U test. Within-group comparisons (T0 vs T2) for continuous variables were carried out by the Wilcoxon signed-rank test. Within-group comparisons for three observation times (T0, T1 and T2) for continuous variables were carried out by the Friedman’s test (non parametric version of repeated measures analysis of variance). A significant result was followed up by post-hoc analysis (Dunn-Sidak criterion) to compare values at T0 vs T1, T0 vs T2 and T1 vs T2. The association between psychological variables, HRQoL and adherence to PAP treatment at 1-year follow-up was assessed by correlation analysis (Spearman’s correlation coefficient). All statistical tests were two-tailed and a p-value <0.05 was considered statistically significant. When appropriate, multiple comparison procedures were used, controlling the false discovery rate (FDR) at 5% using the Benjamini-Hochberg method. All analyses were carried out using the SAS/STAT statistical package, release 9.4 (SAS Institute Inc., Cary, NC, USA)." 

Moreover, in the results paragraph (3.3) we have put the new figure and we corrected as follow: 

"Considering patients with EQ VAS scores available at T0, T1 and T2 (N=47), corresponding scores were 73.6±16.8, 84.8±15.2 and 82.1±18.2. Friedman’s analysis revealed a significant time effect (p=0.0002) and post hoc analysis showed that EQ VAS scores significantly got better at T1 (p=0.0008, T1 vs T0) and this improvement was maintained at T2 (p=0.002, T2 vs T0). Conversely, no difference between T1, and T2 was observed (p=0.97)."